# Outbred Mice with Streptozotocin-Induced Diabetes Show Sex Differences in Glucose Metabolism

**DOI:** 10.3390/ijms24065210

**Published:** 2023-03-08

**Authors:** Boyoung Kim, Eun-Sun Park, Jong-Sun Lee, Jun-Gyo Suh

**Affiliations:** 1Institute of Medical Science, College of Medicine, Hallym University, 1, Hallymdaehak-gil, Chuncheon 24252, Republic of Korea; bykim@hallym.ac.kr; 2Department of Medical Genetics, College of Medicine, Hallym University, 1, Hallymdaehak-gil, Chuncheon 24252, Republic of Korea; gl7sunny@naver.com (E.-S.P.); alex0827@naver.com (J.-S.L.)

**Keywords:** outbred mouse, type 2 diabetes, pancreatic beta cell, glucose metabolism, sex differences

## Abstract

Outbred mice (ICR) with different genotypes and phenotypes have been reported to be more suitable for scientific testing than inbred mice because they are more similar to humans. To investigate whether the sex and genetic background of the mice are important factors in the development of hyperglycemia, we used ICR mice and divided them into male, female, and ovariectomized female (FOVX) groups and treated them with streptozotocin (STZ) for five consecutive days to induce diabetes. Our results show that fasting blood glucose and hemoglobin A_1c_ (HbA_1c_) levels were significantly higher in diabetes-induced males (M-DM) and ovariectomized diabetes-induced females (FOVX-DM) than in diabetes-induced females (F-DM) at 3 and 6 weeks after STZ treatment. Furthermore, the M-DM group showed the most severe glucose tolerance, followed by the FOVX-DM and F-DM groups, suggesting that ovariectomy affects glucose tolerance in female mice. The size of pancreatic islets in the M-DM and FOVX-DM groups was significantly different from that of the F-DM group. The M-DM and FOVX-DM groups had pancreatic beta-cell dysfunction 6 weeks after STZ treatment. Urocortin 3 and somatostatin inhibited insulin secretion in the M-DM and FOVX-DM groups. Overall, our results suggest that glucose metabolism in mice is dependent on sex and/or genetic background.

## 1. Introduction

Rodents, especially mice, are similar to humans at the genetic, anatomical, physiological, and pathophysiological levels, and can therefore replace humans for scientific testing [1]. Laboratory mice are the most commonly used experimental animals for understanding biological functions and translating them to humans in biomedical research [2]. Laboratory mice can be divided into inbred and outbred groups. For decades, inbred mouse strains have been more frequently used than outbred mouse strains because the first, which are genetically identical, have less phenotypic diversity than outbred mice (ICR) [3,4]. Therefore, inbred mice are usually selected for immunological, population genetic mapping, and molecular genetic studies because they are genetically stable, have almost the same phenotype, and have clean genetic background information [5].

Nevertheless, some studies suggest that ICR mice with different genotypes and phenotypes are more similar to humans than inbred mice. In research by Shan Wang et al., it was confirmed that inbred and outbred strains show different responses to metabolic enzymes and agonists [6]. In an inflammatory disease study, ICR mice were found to have greater variability in the phenotypic progression of immune cells compared with inbred mice [7]. These studies suggest that the common cause of the above results is the influence of the different genetic backgrounds of the two strains, which is a key factor to consider in biomedical studies. Outbred strains are more suitable for drug or therapeutic research in humans [8].

In a previous study, we induced diabetes with streptozotocin (STZ) in C57BL/6J mice, which are preferentially used in studies with inbred mouse strains [9]. We confirmed that there were phenotypic differences according to sex in the diabetic groups. For instance, fasting blood glucose (FBG) and hemoglobin A_1c_ (HbA_1c_) levels were significantly higher in male diabetic mice than in female diabetic mice. These results suggest that glucose metabolism is more affected by STZ in male diabetic mice than in female diabetic mice.

In this study, we further investigated whether the sex and genetic background of mice are important factors for the development of hyperglycemia. To test our hypothesis, ICR mice, which have high genetic diversity within the strain, were used to develop hyperglycemia by STZ injection over 5 days. FBG and HbA_1c_ levels were higher in diabetic male mice and ovariectomized diabetic female mice than in diabetic female mice. In addition, pancreatic beta-cell dysfunction was more severe in male diabetic mice than in female diabetic mice. The expression of somatostatin increased in diabetic male and ovariectomized diabetic female mice through delta-cell-mediated regulation. These results demonstrate that glucose metabolism is dependent on sex and/or genetic background in mice.

## 2. Results

### 2.1. STZ-Treated Male Mice Rapidly Developed Hyperglycemia and Diabetes

FBG levels in STZ-treated diabetic male mice (M-DM) were significantly higher than in the M-control group from 1 to 6 weeks (except for weeks 4 and 5) after STZ treatment (Figure 1A). FBG levels were also significantly higher in the FOVX-DM group than in the FOVX-ctrl group at 3–6 weeks (Figure 1B). However, there was no significant difference in FBG levels between the F-DM and F-ctrl groups (Figure 1C). FBG levels increased sequentially in the M-DM, FOVX-DM, and F-DM groups. These results indicate that the FOVX-DM group was more sensitive to STZ treatment than the F-DM group.

At weeks 3 and 6 after STZ treatment, HbA_1c_ levels were significantly higher in the M-DM and FOVX-DM groups than in the control group, whereas the F-DM group had no significant difference from the control group (Figure 1D). In addition, HbA_1c_ levels were significantly higher in the M-DM group than in the F-DM group. HbA_1c_ levels in the FOVX-DM group were also significantly higher than in the F-DM group 6 weeks after STZ treatment. These results show that the FOVX-DM group is more similar to the M-DM than the F-DM group.

### 2.2. STZ-Treated Male Mice Showed the Strongest Glucose Tolerance

The OGTT was performed 3 and 6 weeks after STZ treatment to confirm whether glucose tolerance was induced. At week 3, FBG levels in the M-DM, FOVX-DM, and F-DM groups were significantly higher than those in each control group 30, 60, and 120 min after glucose administration (Figure 2A). Similarly, AUC for blood glucose was significantly higher in all diabetic groups than in each control group at week 3. The AUC in the M-DM and FOVX-DM groups was significantly higher than that in the F-DM group. In addition, the AUC in the M-DM group was significantly higher than that in the FOVX-DM group 3 weeks after STZ treatment (Figure 2B). At week 6, glucose levels showed a similar pattern to that at week 3 (Figure 2C), and the AUC of the M-DM and FOVX-DM groups was significantly higher than that of the F-DM group at week 6. However, there was no significant difference between the M-DM and FOVX-DM groups (Figure 2D). Taken together, these results suggest that ovariectomy affects glucose tolerance in female mice.

### 2.3. The Size of Pancreatic Islets in M-DM and FOVX-DM Was Reduced by STZ Treatment

Morphological evaluation was performed to confirm the size of the pancreatic islets after STZ treatment using hematoxylin–eosin staining. There was no significant difference between the DM and control groups 3 weeks after STZ treatment. In contrast, the size of the pancreatic islets was significantly decreased in the diabetic groups at week 6 compared with the control groups, except in the female group (Figure 3A,B). These results show that the pancreatic islets in the M-DM and FOVX-DM groups were severely damaged over time because they were more affected by STZ treatment.

### 2.4. The Cell Composition of Pancreatic Islets Was Altered in the M-DM and FOVX-DM Groups after STZ Treatment

To identify the cell composition of the pancreatic islets, α- and β-cells were confirmed by immunofluorescence staining. The α-cells, which produce glucagon, are mainly located at the outer edge of the pancreatic islets. The β-cells, producing insulin, are located in the center of the pancreatic islets. The α-cells shifted from the outer border to the center of the pancreatic islets in diabetic mice 6 weeks after STZ treatment (Figure 4A). The number of α- and β-cells in the M-DM and FOVX-DM groups was significantly higher than that in the control group at week 6, whereas the number in the F-DM group was not significantly different from that in the F-ctrl group (Figure 4B). The glucagon/insulin ratios in the M-DM and FOVX-DM groups were significantly higher than in the control group (Figure 4C). These results reveal that the number of α-cells in pancreatic islets was increased by STZ treatment.

### 2.5. Groups M-DM and FOVX-DM Had Impaired Pancreatic Beta-Cell Function

Pancreatic beta-cell function (%B) and insulin sensitivity (%S) were calculated from FBG and plasma insulin levels. The %B of the M-DM and FOVX-DM groups was significantly lower than that of the F-DM group at weeks 3 and 6. Moreover, the %B of the M-DM and FOVX-DM groups was significantly lower than that of the control group at 6 weeks (Figure 5A). The %S in the M-DM group was significantly lower than that in the F-DM group 3 weeks after STZ treatment. The %S of the F-DM group was significantly higher than that of the FOVX-DM group at 6 weeks (Figure 5B). The HOMA-IR in the M-DM and FOVX-DM groups tended to be increased compared with the control groups at weeks 3 and 6 (Figure 5C). Overall, %B and %S were more pronounced in the M-DM group than in the FOVX-DM group.

### 2.6. Insulin Secretion in the M-DM and FOVX-DM Groups Was Inhibited by Urocortin 3 and Somatostatin

To confirm the damage to β-cells induced by STZ treatment, the expression of urocortin 3 (UCN3), a marker of mature β-cells, was measured. In the control groups (M-ctrl, FOVX-ctrl, and F-ctrl), 60–80% of β-cells expressed UCN3 at weeks 3 and 6 after STZ treatment (Figure 6A–C). This ratio was significantly lower in the STZ-treated groups than in the control groups at week 6 (Figure 6C). Somatostatin, produced by pancreatic delta cells, prevents the secretion of pancreatic hormones such as insulin and glucagon. The intensity of somatostatin was significantly increased in the M-DM and FOVX-DM groups com-pared with the control group at week 6. These results indicate that the secretion of insulin and glucagon was inhibited in the M-DM and FOVX-DM groups compared with the F-DM group (Figure 6D).

## 3. Discussion

Type 2 diabetes is a disorder of glucose metabolism, with a metabolic phenotype that differs according to sex [10]. Our group reported that male inbred mice (C57BL/6J) had higher hyperglycemia than female mice after STZ injection [9]. In this study, we investigated whether the genetic background of the mice was important for the development of hyperglycemia and whether female hormones such as estrogen play a role in glucose metabolism after STZ injection. To test our hypothesis, ICR mice, which have high genetic diversity within the strain, were used to develop hyperglycemia by STZ injection over 5 days.

FBG and HbA_1c_ levels were significantly increased in the M-DM and FOVX-DM groups compared with the control groups, whereas the F-DM group showed no significant difference compared with the control group. Additionally, the levels of M-DM and FOVX-DM were significantly higher than those of F-DM (Figure 1). In this study, all DM groups developed glucose tolerance, regardless of sex. Notably, glucose tolerance in the M-DM group was significantly different from that in the FOVX-DM and F-DM groups 3 weeks after STZ treatment. At 6 weeks, glucose tolerance in the M-DM group was not significantly different from that in the FOVX-DM group, indicating that ovariectomy might affect glucose tolerance in female mice (Figure 2). These results indicate that female ICR mice did not exhibit hyperglycemia under the same conditions in which male mice developed diabetes. In addition, the phenotype of glucose metabolism in ovariectomized female mice was similar to that of male mice. Our results show that female hormones are a principal factor in the development of hyperglycemia after STZ injection. In contrast, Kim et al. [9] reported that inbred female mice (C57BL/6J) developed hyperglycemia under the same conditions. These different results could be due to genetic differences between the inbred and outbred mice.

The size of the pancreatic islets was significantly smaller in the M-DM and FOVX-DM groups than in the control group (Figure 3). In the M-DM and FOVX-DM groups, the intensities of α- and β-cells were significantly increased compared with the intensity observed in each control group at week 6 after STZ treatment, whereas it was not significantly different in the F-DM group compared with the F-ctrl group. The glucagon/insulin ratio was significantly higher in the diabetic group than in the control group, except in the female group (Figure 4). The β-cell functions were significantly decreased in the M-DM and FOVX-DM groups compared with the control group at week 6 after STZ treatment. Insulin sensitivity was significantly lower in the FOVX-DM group than in the F-DM group (Figure 5). Remedi et al. [11] reported that pancreatic cells changed due to glucose toxicity when hyperglycemia was continuously maintained. Therefore, the size of pancreatic islets in M-DM and FOVX-DM groups decreased due to the death of β-cells, indicating that cell function deteriorated. Taken together, the morphological characteristics of pancreatic islets in M-DM, FOVX-DM, and F-DM were consistent with the results of diabetic parameters such as hyperglycemia and glucose tolerance.

The β-cells expressing urocortin 3 (UCN3) were significantly decreased in the M-DM, FOVX-DM, and F-DM groups compared with the control groups at week 6 after STZ treatment. In addition, the intensity of somatostatin was significantly increased in the M-DM and FOVX-DM groups compared with the F-DM group at week 6, indicating that the insulin secretion was inhibited in the M-DM and FOVX-DM groups compared with control groups (Figure 6). UCN3 is expressed in murine β-cells, but not in α- and δ-cells [12]. UCN3 is the most appropriate marker for β-cell maturation [13,14,15]. In this study, the expression of UCN3 was significantly lower in the diabetic group than in the control group. This result suggests that the number of functionally mature β-cells was reduced by STZ injection. UCN3 stimulates somatostatin secretion in δ-cells [16] and regulates the negative feedback of somatostatin to decrease insulin secretion [17]. As the expression of the somatostatin was significantly decreased in the diabetic groups, the intensity of glucagon and insulin was significantly increased in the pancreatic islets of these groups, but not in the female diabetic group. Researchers have reported on how estrogen protects the beta cells of the pancreas from damage caused by STZ [18,19]. Overall, these results suggest that female hormones play an important role in the regulation of glucose metabolism in the STZ-induced diabetic mouse model.

In conclusion, ICR mice showed different glucose metabolism characteristics compared with inbred mice (C57BL/6J) that, under the same conditions, developed diabetes after STZ injection. In addition, female hormones play an important role in regulating glucose metabolism, regardless of the genetic background of the mice. These results provide important information for the development of animal models and efficacy testing of drugs or functional foods for the treatment of diabetes.

## 4. Materials and Methods

### 4.1. Experimental Animals and Streptozotocin (STZ) Treatment

Five-week-old ICR mice were obtained from Orientbio (Seoul, Republic of Korea) and had an adaptation period of 1 week. The mice were fed a chow diet (Purina, Seongnam-si, Republic of Korea) and water ad libitum. At 6 weeks of age, ovariectomy was performed to establish a postmenopausal female mouse model, followed by a recovery period of 2 weeks. Mice were divided into the following six groups: (1) male control (M-ctrl, *n* = 10), (2) STZ-treated diabetic males (M-DM, *n* = 12), (3) female control (F-ctrl, *n* = 10), (4) STZ-treated diabetic females (F-DM, *n* = 12), (5) ovariectomized female control (FOVX-ctrl, *n* = 15), and (6) ovariectomized STZ-treated diabetic females (FOVX-DM, *n* = 16). To induce type 2 diabetes, mice were continuously treated with STZ (40 mg/kg, Sigma, St. Louis, MO, USA) in 50 mM citrate buffer (pH 4.5) by intraperitoneal injection for 5 days at 8 weeks of age [9,20]. During STZ administration, mice were simultaneously administered a 10% sucrose solution to prevent hypoglycemic shock. Mice were reared in an animal care facility at a temperature of 22 ± 2 °C, a humidity of 55 ± 10%, and a 12 h light–dark cycle (8 am–8 pm).

### 4.2. Measurement of FBG and HbA_1c_

To measure FBG levels from 0 to 6 weeks after STZ treatment, mice were fasted for 6 h, and blood was collected from the retro-orbital plexus using a heparin capillary tube. FBG levels were measured using blood glucose meters (ACCU-CHEK; Roche, Indianapolis, IN, USA). Hemoglobin A_1c_ levels, indicating the average blood glucose level over the past 2–3 months, were measured at 0, 3, and 6 weeks after STZ treatment. Whole blood collected in a heparin tube (BD Microtainer, Franklin Lakes, NJ, USA) was analyzed using an HbA_1c_ analyzer (CareSURETM Analyzer 100, Wellsbio, Seoul, Republic of Korea) and dedicated cartridges (CareSURETM A_1c_ cartridge, Wellsbio, Seoul, Republic of Korea).

### 4.3. Oral Glucose Tolerance Test (OGTT) and Area under the Curve (AUC)

To confirm whether glucose tolerance was induced in the diabetic mice, an OGTT was performed at weeks 3 and 6. All mice were fasted for 6 h and received an oral gavage injection of 2 g/kg *D*-glucose (Sigma-Aldrich, St. Louis, MO, USA) in saline. Blood glucose was measured using blood glucose meters (ACCU-CHEK, Roche, Indianapolis, IN, USA) at 0, 30, 60, and 120 min after *D*-glucose treatment. Then, the AUC was calculated as the geometric mean value based on the area under the OGTT curve.

### 4.4. Histological Analysis of the Pancreas

After the experiment, mice were anesthetized with isoflurane (flowmeter, 100–200 mL/min, vaporizer 2–3%) and subjected to transcardiac perfusion with 4% paraformaldehyde. The pancreas was dissected, embedded in paraffin blocks, and cut into 5 µm slices with a microtome (Reichert-Jung 2030 Biocut, Nußloch, Germany). To determine the size of the pancreatic islets, images were captured with an optical microscope (Diaphot 300, Nikon, Tokyo, Japan). The size was measured using Image J software version 1.8.0_172 (NIH, Stapleton, NY, USA).

For immunofluorescence evaluation, slides were blocked with 10% horse serum in Tris-buffered saline (TBS) and stained with anti-insulin (1:200, Invitrogen, Waltham, MA, USA), anti-glucagon (1:200, Cell Signaling, Denver, MA, USA), anti-somatostatin (1:100, Invitrogen, Waltham, MA, USA), and anti-urocortin III (1:200, Phoenix Pharmahandel, Mannheim, Germany) overnight at room temperature. On the second day, slides were washed with TBS and incubated with donkey anti-guinea pig Alexa Fluor 594 (1:200; Jackson ImmunoResearch, West Grove, PA, USA), donkey anti-rabbit Alexa Fluor 488 (1:100; Abcam, Cambridge, UK), and donkey anti-rat Alexa Fluor 647 (1:200; Abcam, Cambridge, UK) antibodies. After staining, islets were imaged using confocal microscopy (LSM 710, Carl Zeiss, Jena, Germany), and the intensities of insulin, glucagon, and somatostatin were measured using ZEISS Blue software (version 11).

### 4.5. Homeostatic Model Assessment for Insulin Resistance (HOMA-IR)

Insulin levels were measured in all plasma samples and analyzed using a microplate reader (VersaMax^TM^, Molecular Devices, San Jose, CA, USA). A public HOMA2 calculator (https://www.dtu.ox.ac.uk/homacalculator/ accessed on 15 March 2022) was used to assess pancreatic beta-cell function (%B), insulin sensitivity (%S), and insulin resistance.

### 4.6. Statistical Analysis

All data are expressed as mean ± standard error of the mean. Normally distributed data were tested for homogeneity of variance using Levene’s test. Data were analyzed using an independent two-sample *t*-test when the data followed an equal variance. Statistical significance was set at * *p* < 0.05, ** *p* < 0.01, *** *p* < 0.001, and **** *p* < 0.0001. All results were analyzed using IBM SPSS Statistics 25 (SPSS Inc., Chicago, IL, USA).

## Figures and Tables

**Figure 1 ijms-24-05210-f001:**
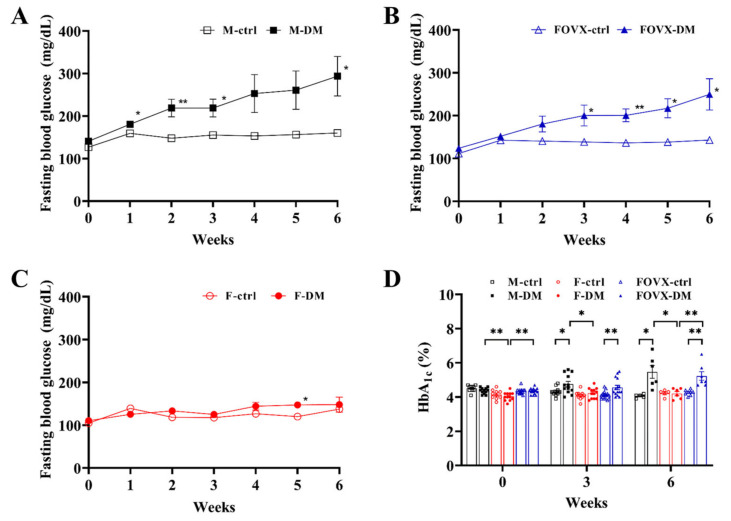
Changes in fasting blood glucose (FBG) and hemoglobin A_1c_ (HbA_1c_) levels in STZ-treated diabetic mice over 6 weeks. FBG levels of male (**A**), ovariectomized female (**B**), and female (**C**) groups. (**D**) HbA_1c_ levels at 3 and 6 weeks after STZ treatment. FBG and HbA_1c_ levels of FOVX-DM mice were significantly increased compared with those of F-DM mice. Data were analyzed using an independent two-sample *t*-test. * *p* < 0.05, ** *p* < 0.01 compared with the respective control group.

**Figure 2 ijms-24-05210-f002:**
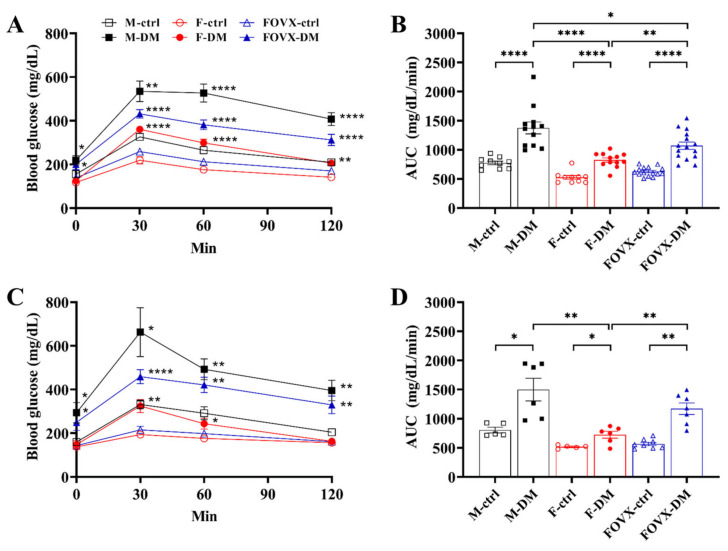
Results of oral glucose tolerance test (OGTT) in STZ-treated diabetic mice. OGTT results and area under the curve (AUC) at weeks 3 (**A**,**B**) and 6 (**C**,**D**) after STZ treatment. Glucose tolerance of the FOVX-DM group was worse at week 6 than at week 3, indicating that the ovaries play an important role in glucose metabolism. Data were analyzed using an independent two-sample *t*-test. * *p* < 0.05, ** *p* < 0.01, **** *p* < 0.0001 compared with the respective control group.

**Figure 3 ijms-24-05210-f003:**
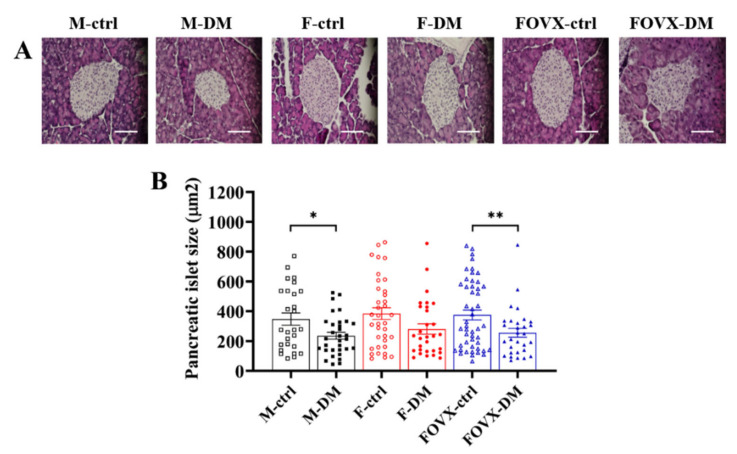
Pancreatic islets’ size at week 6 after STZ treatment. Hematoxylin and eosin staining (**A**) and mean value of pancreatic islets (**B**) in all groups. The size of pancreatic islets was significantly decreased in the M-DM and FOVX-DM mice compared with the control mice. Data were analyzed using an independent two-sample *t*-test. * *p* < 0.05, ** *p* < 0.01 compared with the respective control group.

**Figure 4 ijms-24-05210-f004:**
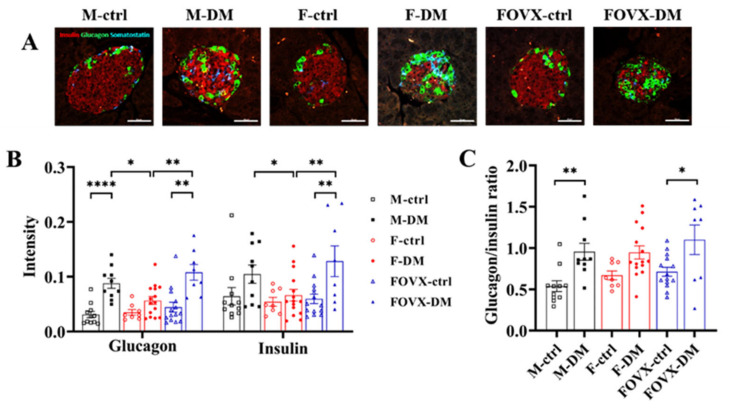
Differences in cell composition between the diabetic and control groups at week 6 after STZ treatment. (**A**) Immunofluorescence staining of pancreatic islets was performed with anti-glucagon and anti-insulin. The intensities of α-cells (glucagon, green) and β-cells (insulin, red) (**B**) and the glucagon/insulin ratio (**C**) were measured. The α-cells of M-DM and FOVX-DM were significantly increased compared with each control group. Data were analyzed using an independent two-sample *t*-test. * *p* < 0.05, ** *p* < 0.01, **** *p* < 0.0001 compared with the respective control group.

**Figure 5 ijms-24-05210-f005:**
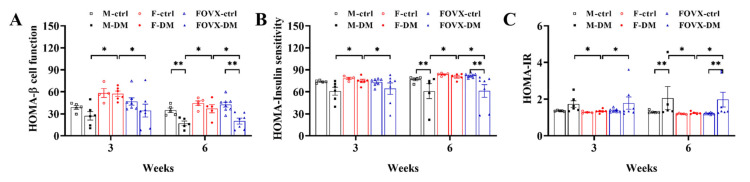
Changes in pancreatic beta-cell function (HOMA-β cell function) and insulin sensitivity (HOMA-Insulin sensitivity) in diabetic groups. Beta-cell function (**A**), insulin sensitivity (**B**), and HOMA-IR (**C**) were determined at 3 and 6 weeks using the public HOMA2 calculator. The beta-cell function of M-DM and FOVX-DM was significantly lower than that of F-DM. Data were analyzed using an independent two-sample *t*-test. * *p* < 0.05, ** *p* < 0.01 compared with the respective control group.

**Figure 6 ijms-24-05210-f006:**
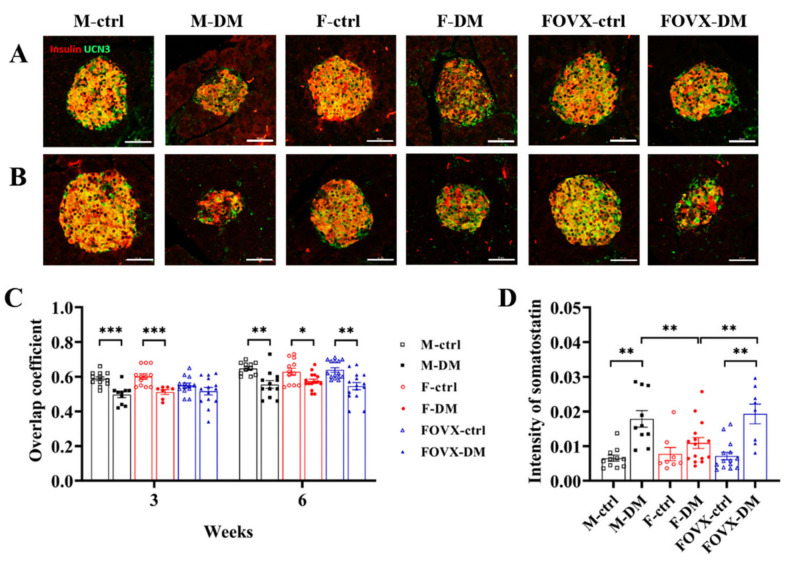
Differences in the expression of UCN3 and somatostatin between diabetic and control groups. To detect the expression of UCN3, immunofluorescence staining was performed at weeks 3 (**A**) and 6 (**B**). The overlap coefficient of UCN3 and insulin (**C**) was evaluated. The intensity of somatostatin (**D**) was measured at week 6. The cells expressing both UCN3 and insulin were significantly reduced in the diabetic groups compared with the control groups at week 6. The overlap coefficient indicates the association between the two distributions by the regions where the expression of the two markers (insulin, UCN3) overlap. Data were analyzed using an independent two-sample *t*-test. * *p* < 0.05, ** *p* < 0.01, *** *p* < 0.001 compared with the respective control group.

## Data Availability

Not applicable.

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
