# Peer review of "Outbred Mice with Streptozotocin-Induced Diabetes Show Sex Differences in Glucose Metabolism"

_ijms, 2023, doi:10.3390/ijms24065210_

Round 1

Reviewer 1 Report

1. The dose of the STZ seems to be pretty high for 5 days. Usually a single dose of STZ (65 mg/kg) is sufficient to induce diabetes. 40 mg/kg X5 days=200 mg/kg seems to be lethal dose. what about the mortality? 

2. what about the ketosis in these mice?

3. The possible mechanisms of observed changes are not discussed properly.

4. Sex is a major determinant in the outcomes of many diseases. Many studies showed sex-specific differences in many autoimmune diseases including  Type 1 diabetes (T1D). STZ induced diabetes mimics T1D. Discussion needs to be revised.

5. Figure 5A-B, Y-axis labels are not clear.

6. Explain about overlap coefficient in figure legend (Fig. 6C)

Author Response

We have addressed to all the comments suggested by the reviewer, and implemented the corresponding changes directly in the revised manuscript according to the comments and where appropriate. Please find point-by-point detailed responses to the reviewer’ comments.

1. The dose of the STZ seems to be pretty high for 5 days. Usually a singe dose of STZ (65mg/kg) is sufficient to induce diabetes. 40mg/kg X 5 days = 200mg/kg seems to be lethal dose. What about the mortality?

 -> A single high-dose (200 mg./kg) STZ injection causes type I diabetes and death within 10 days after injection. In this study, type 2 diabetes was induced by intraperitoneal injection of STZ (40 mg/kg) for a period of 5 days. This is a well established method (Furman et al, 2021, Kim et al., 2020). In addition, the mortality rate in the same condition is 0%.

Furman BL. Streptozotocin-Induced Diabetic Models in Mice and Rats. Curr Protoc (2021) 1(4):e78. doi: 10.1002/cpz1.78.

Kim B, Kim YY, Nguyen PTT, et al. Sex differences in glucose metabolism of streptozotocin-induced diabetes inbred mice (C57BL/6J). Appl Biol Chem 63, 59 (2020).

2.  What about the ketosis in these mice?

-> Ketosis is a metabolic condition with elevated levels of ketone bodies in the blood or urine. The ketosis occurs during the conditions of low availability of glucose in the body. We did not measure ketones in our study because mice with type 2 diabetes were in a hyperglycemic state.

3.  The possible mechanisms of observed changes are not discussed properly.

-> Our findings suggest that estrogen may protect against STZ-induced beta cell damage in the pancreas. In addition, Le May et al, have been reported how estrogen protects pancreatic beta cell damage from STZ (May, et al., 2006). We have added a few sentences to the discussion.

May CL, Chu K, Hu M, et al. Estrogens protect pancreatic beta-cells from apoptosis and prevent insulin-deficient diabetes mellitus in mice. Proc Natl Acad Sci U S A. 103, 9232-9237 (2006).

4. Sex is a major determinant in the outcomes of many diseases. Many studies showed sex-specific differences in many autoimmune diseases including Type 1 diabetes (T1D). STZ induced diabetes mimics T1D. Discussion needs to be revised.

-> Beta cells are destroyed by the immune system (autoimmune disease) in the type 1 diabetes. We used STZ to induce type 2 diabetes in this study. The STZ destroyed the beta cells in the pancreas using nitric oxide (NO). Therefore, we believe that the immune system is not involved in our diabetic mice.

5. Figure 5A-B, Y-axis labels are not clear.

-> Thank you for your comment. In response to your comment, we have changed the Y-axis labels in Figure 5A-B.

6. Explain about overlap coefficient in figure legend (Fig. 6C)

-> We have added a sentence in the legend of Figure 6 to explain the overlap coefficient in response to your comment.

The overlap coefficient indicates the association between the two distributions by the regions where the expression of the two markers (insulin, UCN3) overlap.

Reviewer 2 Report

In this manuscript, Kim et al., demonstrating how sex difference affects the glucose metabolism by streptozotocin induced DM. This is an interesting manuscript, which is overall very well written, however, this article doesn’t have novelty and lack of experiments to prove their conclusion. It is very well known that estrogen has protective effect on DM and authors are not providing the data for Inbred mice.

Major Comments

1.     Most of the results doesn’t have significant difference between the groups to demonstrate the sex difference in the glucose metabolism and even changes are minimal.

2.     One of the major drawbacks of this manuscript is, authors doesn’t provide enough data to support their conclusion. Must include inbred mice for the comparison with outbred.

3.     Figure 3. is not clear and there is no difference in the pancreatic islets between F-Ctrl and F-DM. Author must do experiments to prove that how estrogen beneficial against STZ induce DM.

4.     Methods section is not clear especially STZ injection. Why authors continuously injected STZ for induce DM and this may alter the results., since one-time 40mg/Kg injection is enough to induce DM.

5.     I would like to change “oral injection” to “by oral gavage needle.”

Author Response

We have addressed to all the comments suggested by the reviewer, and implemented the corresponding changes directly in the revised manuscript according to the comments and where appropriate. Please find point-by-point detailed responses to the reviewer’ comments.

1. Most of the results doesn’t have significant difference between the groups to demonstrate the sex difference in the glucose metabolism and even changes are minimal.

-> In this study, although mice were injected with the same dose of STZ, the fasting blood glucose results were significantly higher in the order of M-DM > FOVX-DM > F-DM. Glucose tolerance by OGTT was also shown in the same order. Taken together, these results suggest that glucose metabolism differs between the sexes.

2. One of the major drawbacks of this manuscript is, authors doesn’t provide enough data to support their conclusion. Must include inbred mice for the comparison with outbred.

-> Our previous study reported that inbred (C57BL/6J) female mice developed diabetes under the same conditions used here (Kim et al, 2020). However, outbred (ICR) female mice did not show hyperglycemia under the same conditions in this study. We believe that this difference is due to the genetic background between inbred and outbred mice. We mentioned this part in the discussion.

Kim B, Kim YY, Nguyen PTT, et al. Sex differences in glucose metabolism of streptozotocin-induced diabetes inbred mice (C57BL/6J). Appl Biol Chem 63, 59 (2020).

3.  Figure 3. is not clear and there is no difference in the pancreatic islets between F-Ctrl and F-DM. Author must do experiments to prove that how estrogen benefical against STZ induced DM.

-> The mean size of pancreatic islets was significantly reduced in both male and ovariectomized female mice when injected with the same dose of STZ, but not in estrogen-secreting females. This finding suggests that estrogen may protect against STZ-induced beta cell damage in the pancreas. In addition, Le May et al, (2006) have been reported how estrogen protects pancreatic beta cell damage from STZ.

May CL, Chu K, Hu M, et al. Estrogens protect pancreatic beta-cells from apoptosis and prevent insulin-deficient diabetes mellitus in mice. Proc Natl Acad Sci U S A. 103, 9232-9237 (2006).

4. Methods section is not clear especially STZ injection. Why authors continuously injected STZ for induce DM and this may alter the results, since one-time 40mg/kg injection in enough to induce DM.

-> A single high-dose (40 mg./kg) injection of STZ could cause hyperglycemis for a short period of time and will be restored naturally. In this study, type 2 diabetes was induced by intraperitoneal injection of STZ (40 mg/kg) for a period of 5 days. This is a well established method (Furman et al, 2021, Kim et al., 2020).

Furman BL. Streptozotocin-Induced Diabetic Models in Mice and Rats. Curr Protoc (2021) 1(4):e78. doi: 10.1002/cpz1.78.

Kim B, Kim YY, Nguyen PTT, et al. Sex differences in glucose metabolism of streptozotocin-induced diabetes inbred mice (C57BL/6J). Appl Biol Chem 63, 59 (2020).

Round 2

Reviewer 1 Report

Authors have revised the manuscript as per reviewer's suggestion

Author Response

Authors have revised the manuscript as per reviewer's suggestion

-> Thank you for your comments to improve the quality of our paper.

Reviewer 2 Report

Authors has to state their comments in the manuscript especially the need of continuous injection of STZ since there are ample of publications showing one time injection is enough to induce DM.

Author Response

We have revised the manuscript according to the comment. The following is a point-by-point response to the reviewer’s comment.

Authors has to state their comments in the manuscript especially the need of continuous injection of STZ since there are ample of publications showing one time injection is enough to induce DM.

-> Thank you for your comment. In Materials and Methods, we cited papers showing conditions that cause type 2 diabetes from STZ (Furman et al. 2012, Kim et al., 2020).

Furman BL. Streptozotocin-Induced Diabetic Models in Mice and Rats. Curr Protoc (2021) 1(4):e78. doi: 10.1002/cpz1.78.

Kim B, Kim YY, Nguyen PTT, et al. Sex differences in glucose metabolism of streptozotocin-induced diabetes inbred mice (C57BL/6J). Appl Biol Chem 63, 59 (2020).
